# Fish Welfare-Related Issues and Their Relevance in Land-Based Olive Flounder (*Paralichthys olivaceus*) Farms in Korea

**DOI:** 10.3390/ani14111693

**Published:** 2024-06-05

**Authors:** Seoyeon Oh, Seunghyung Lee

**Affiliations:** 1Fisheries Policy Research Department, The Korean Maritime Institute Busan, Busan 49111, Republic of Korea; seoyeon_oh@kmi.re.kr; 2Major of Aquaculture and Applied Life Sciences, Division of Fisheries Life Sciences, Pukyong National University, Busan 48513, Republic of Korea

**Keywords:** animal welfare, aquaculture, interviews, one health, surveys

## Abstract

**Simple Summary:**

The welfare of farmed fish is crucial in aquaculture, requiring demanding attention to ensure their health and safety throughout the farming process. Stress and pain experienced by fish have profound implications for their overall well-being, highlighting the importance of mitigating these factors. Enhancing fish welfare promotes the sustainability of aquaculture and fortifies fish resilience against stressors. Despite significant advancements in production and species diversity in Korean aquaculture, the welfare of farmed fish remains largely neglected. To address potential welfare issues in the farming process, we conducted an analysis focusing on olive flounder, a staple species in Korean aquaculture. We identified welfare concerns relevant to olive flounder farming, laying the foundation for the development of robust welfare standards and the promotion of ethical and responsible practices in Korean aquaculture.

**Abstract:**

Korean aquaculture has expanded considerably in recent decades; however, this growth has often prioritized quantity over fish welfare. Therefore, we analyzed the aquaculture practices of olive flounder, the predominant species in Korean consumption and production, within the framework of fish welfare. We conducted extensive interviews and surveys across olive flounder farms in Jeju-do and Wando to examine prevalent issues impacting fish welfare in aquaculture. These issues include stressors, mass mortality events, and disease outbreaks, all of which strain the welfare of farmed fish. Moreover, our survey revealed farmers’ varying perceptions of fish welfare, highlighting the necessity for a cohesive approach. Accordingly, we propose recommendations to enhance fish welfare and establish a more sustainable aquaculture model in Korea. Ensuring fish welfare in aquaculture operations requires a comprehensive approach that considers the physiological and behavioral needs of fish throughout the farming lifecycle. By prioritizing fish welfare, Korean aquaculture can strengthen its growth while maintaining ethical standards and ensuring the well-being of farmed fish. This welfare-centric approach is crucial for the long-term sustainability and resilience of the Korean aquaculture industry. By addressing welfare concerns and promoting responsible practices, Korean aquaculture can foster an ethically sound and sustainable future.

## 1. Introduction

Fish, similar to other animals, demonstrate sentience and are capable of experiencing pain, fear, anxiety, and stress [1,2,3,4,5,6,7], exhibiting sensory receptors sensitive to pain and physiological and behavioral stress responses similar to mammals when subjected to painful handling procedures [4,5,6,7,8,9,10,11,12,13,14,15]. Thus, vigilantly monitoring fish welfare is essential to ensure their humane treatment and a high quality of life free from pain and suffering [4,5,6,7,8,9,10,11,12,13,14,15,16,17].

While a universal definition of fish welfare remains unresolved, the World Organization for Animal Health (WOAH) considers it crucial from a One Health perspective [18], with fish welfare protocols increasingly recognized as pivotal within animal welfare for guaranteeing humane treatment across fish farming activities, including transportation and slaughter. The European Union has pioneered legislative initiatives and policies prioritizing animal welfare, exemplified by Council Directive 98/58 in July 1998, aimed at safeguarding animals, including fish raised for farming [19], marking a significant milestone in animal welfare legislation.

Norway prioritizes fish health and welfare, implementing comprehensive standards throughout the fish farming continuum to address diseases, parasitic infections, handling, injuries, and mortality during farming and transportation [20,21,22,23]. Conversely, Korean aquaculture has experienced substantial growth over the past three decades [24]. However, emphasizing quantitative production has led to excessive inputs, elevated mortality rates, food safety challenges, and environmental degradation [24,25,26,27].

Integrating fish welfare considerations throughout the aquaculture process is essential to ensure the sustainability of fish farming and mitigate issues arising from conventional quantitative production methods. Prioritizing aquaculture methodologies that meticulously consider fish stress and suffering is a crucial step toward addressing this concern, thereby safeguarding farmed fish welfare and reducing adverse environmental impacts. Therefore, in this study, we aimed to assess current olive flounder (*Paralichthys olivaceus*) farms, which account for the highest production volume (46,000 tons, 50.5% of the total in 2022) among farmed species in Korea [28]. Given the absence of fish welfare guidelines in Korea [29,30], we aimed to identify the most vulnerable aspects of fish welfare in olive flounder farming facilities, including less conspicuous ones, to gather insights informing the development of future fish welfare guidelines. Through field surveys, we aimed to identify prevalent characteristics and challenges in fish farms while assessing the awareness of fish welfare among farmers and their willingness to adopt fish welfare measures. By analyzing olive flounder farming in Korea from a fish welfare perspective, we aimed to determine strategies and measures to ensure the sustainability of flounder farming by addressing relevant welfare issues.

## 2. Materials and Methods

### 2.1. Study Design and Data Collection

In this study, we surveyed 71 fish farms, 36 of which were located in Wando and 35 in Jeju-do, the primary region for olive flounder farming in South Korea (Figure 1). Of the 71 fish farms, 42 had been in operation for over 20 years, while 29 had been in operation for less than 20 years. While olive flounder farming has recently expanded in Pohang, Gyeongsangbuk-do Province, and Gyeongsangnam-do Province (including certain areas in Busan and Geoje Island), it continues on a smaller scale than that in Jeju-do and Wando [28,31,32,33,34,35,36]. Therefore, we limited the survey to the latter two specified regions.

To ensure the accuracy of our findings, we developed and used structured questionnaires (see Appendix A) for face-to-face interviews conducted during the survey period from 1 to 15 September 2023 [37]. Questionnaires were developed based on information obtained from a literature review [38] and expert interviews and designed to allow respondents to indicate yes, no, and do not know. Seventy-one respondents were olive flounder farm representatives, 69 of whom were also involved in farming (see Table 1).

We investigated various aspects of aquaculture production, such as fish mortality and disease causes, feeding management, disease control, fish handling, transportation, awareness of fish welfare among farmers, and welfare-oriented aquaculture practices. The maximum allowable margin of error at a 95% confidence level was ±11.63%. In this survey, we set the target to 71 olive flounder farms in Wando and Jeju-do. Since we did not know the total number of olive flounder farms in these two regions, the sampling error was 11.63% if we set the sample size to an infinite population (100,000) (see Equation (1)). However, if we set the number of olive flounder farms in Wando and Jeju-do to 500, the sampling error was 10.78%.
(1)$$e=Z\times \sqrt{\frac{P\times \left(1-P\right)}{n}}\times \sqrt{\frac{N-n}{N-1}}$$Sampling error formula, where *N* is the population size, *n* is the sample size, *e* is the margin of error or confidence interval, *Z* is the confidence level, and *P* is the observed percentage. Source: cited from https://www.nownsurvey.com/calculator/margin_error (accessed on 5 February 2024).

This study evaluated the regional differences in the awareness of fish welfare, the consideration of fish pain and stress, and the inclination toward welfare-oriented aquaculture practices among aquaculture facilities in Wando and Jeju-do. Through chi-square testing of survey data [39], we aimed to identify significant disparities that could inform tailored educational and policy-making initiatives in the olive flounder aquaculture industry.

### 2.2. Statistical Analysis

We analyzed the differences in responses between Wando and Jeju for each factor using chi-squared tests. The chi-squared test is a statistical method used to determine whether there is a significant association between two categorical variables. The data from the surveys were organized into contingency tables for each factor, with missing values handled appropriately and incomplete data entries excluded from the analysis. For each contingency table, the chi-squared statistic, *p*-value, degrees of freedom, and expected frequencies were calculated using the chi2_contingency function from the scipy.stats library in Python. The chi-squared statistic measures the discrepancy between observed and expected frequencies, and a *p*-value of less than 0.05 was considered statistically significant. Degrees of freedom were determined based on the number of categories in the contingency table. The results were interpreted to identify significant differences in responses between Wando and Jeju-do for each factor, indicating different impacts or perceptions in the two regions. Ethical considerations were maintained by collecting and analyzing data anonymously to ensure the privacy and confidentiality of respondents. The survey was conducted with informed consent, and participants were assured that their responses would only be used for research purposes. The null and alternative hypotheses for each factor analyzed are as follows:
**H0:** *There is no significant difference in the responses between Wando and Jeju-do*.**H1:** *There is a significant difference in the responses between Wando and Jeju-do*.

## 3. Results

### 3.1. Survey Findings on Fish Welfare Issues

#### 3.1.1. Water-Quality Changes

A total of 84.5% of respondents reported changes in water quality as the most significant factor contributing to olive flounder mortality. The chi-squared test result showed a chi-squared statistic of 15.3526 with a *p*-value of 0.0015, indicating a significant difference in responses between Wando and Jeju-do. This suggests that perceptions or experiences related to water quality changes vary significantly between these two regions. 

#### 3.1.2. Water Temperature Variations

A total of 81.7% of the respondents identified high and low water temperatures as a critical factor affecting olive flounder mortality. The chi-squared statistic was 14.9386, with a *p*-value of 0.0019, also indicating a significant difference. Temperature variations are perceived or experienced differently in Wando and Jeju-do, possibly due to regional climatic differences or specific local conditions affecting aquaculture practices.

#### 3.1.3. Parasite and Virus Issues

A total of 80.3% of respondents considered parasites and viruses significant. However, the chi-squared statistic was 3.9422 with a *p*-value of 0.2678, indicating no significant difference between Wando and Jeju-do. This suggests that both regions face similar challenges regarding parasites and viruses or that their management practices are comparable. 

#### 3.1.4. Feed Management

The survey of olive flounder farms showed that 93% of respondents preferred moist pellet (MP) feeds to formulated feeds. Excessive or deteriorated feed was noted by 69% of respondents as a contributing factor to mortality. The chi-squared statistic was exceptionally high at 44.9948, with a *p*-value of 9.2761 × 10^−10^, signifying a highly significant difference in responses. This indicates that the types and amounts of feed used differ greatly between Wando and Jeju-do, which could be influenced by regional practices, feed availability, or preferences. The survey also highlighted that 97% of respondents maintained records to ensure feed quality, and all respondents managed temperatures to prevent feed spoilage. Despite this, overfeeding and feeding spoiled feed were major concerns.

#### 3.1.5. Disease Management in Production

The use of medication was reported as a significant factor, with a chi-squared statistic of 22.1591 and a *p*-value of 2.5097 × 10^−6^. This significant difference indicates that the use of medication in aquaculture is significantly different between Wando and Jeju-do, possibly due to differences in disease prevalence, regulatory environments, or farming practices. The survey results showed that 74.6% of farms frequently used pharmaceuticals, with all Wando respondents confirming frequent use, whereas Jeju-do had a more varied response.

#### 3.1.6. Handling 

All respondents indicated that they use appropriate methods to minimize physical damage and manage fish stress to prevent mortality and produce high-quality fish. The chi-squared test results showed no significant difference between Wando and Jeju-do, with a chi-squared statistic of 0.0 and a *p*-value of 1.0. This indicates complete agreement across both regions, suggesting that both Wando and Jeju-do are equally committed to proper handling practices to ensure fish welfare and quality in aquaculture.

#### 3.1.7. Transportation Practices

Injury during transportation was found to be a significant factor, with a chi-squared statistic of 59.6859 and a *p*-value of 1.1127 × 10^−14^. This suggests a major discrepancy in how injuries during transportation are perceived or occur between Wando and Jeju-do, which could be due to differences in transportation methods, distances, or handling practices. The survey revealed that 53.5% of respondents reported fish injuries during transport, with all Wando respondents indicating transport as a cause of damage, while 88.6% of Jeju respondents reported no injuries.

#### 3.1.8. Workforce Training

The survey results highlight significant regional differences in training practices between Wando and Jeju-do. A total of 71.4% of respondents reported sporadic training sessions in Jeju-do, while in Wando, 91.7% reported regular training sessions conducted 12 times a year or less. The chi-squared test (χ^2^ = 11.29, *p* = 0.0234) confirmed a statistically significant difference in the provision of professional training between the two regions, indicating varied perceptions of its importance. While 83.1% of all respondents provided training, the frequency and duration varied regionally. Jeju-do’s training was more sporadic, whereas Wando’s was more regular but shorter.

### 3.2. Awareness of Fish Welfare 

During our investigation, we observed a noticeable disparity in the awareness of fish welfare among fish farmers across Wando and Jeju-do. Despite the general acknowledgment of the importance of considering fish pain and stress, where 95.8% of respondents recognized its significance in production practices, only 40.8% of the respondents overall confirmed being aware of fish welfare issues. In Wando, 47.2% of respondents reported awareness of fish welfare compared with 34.3% in Jeju-do. However, our chi-square analysis reveals that this difference was not statistically significant (χ^2^ = 0.752, *p* > 0.05), suggesting that awareness of fish welfare is similarly distributed among respondents in both regions. This finding aligns with the recognition of the necessity to consider fish pain and stress, where no significant regional differences were detected (χ^2^ = 0.00062, *p* > 0.05), indicating a consistent perception across the regions.

Conversely, attitudes toward transitioning to welfare-oriented aquaculture practices showcased significant regional variation. The chi-squared test indicated a statistically significant difference between the regions (χ^2^ = 26.92, *p* < 0.05), with only 32.4% of respondents in Wando and Jeju-do combined showing willingness to make such transitions. Notably, 63.4% of respondents were reluctant to adopt welfare-oriented practices, citing “lack of interest” and “not feeling the necessity” as the main reasons. This suggests a pressing need for targeted educational initiatives and policy adjustments to foster a broader acceptance and implementation of welfare-oriented aquaculture practices. These findings underscore the need for more focused efforts to bridge the gap between current practices and the potential benefits of adopting fish welfare measures, which could be enhanced by developing and disseminating targeted educational programs and regulatory incentives designed to promote sustainable practices across these regions.

Producer surveys identified several fish welfare issues related to olive flounder farms (Table 2). The most commonly raised issues were water quality, feed management, disease management, transportation, and training. 

## 4. Discussion

In this study, we examined olive flounder aquaculture facilities and explored producers’ perspectives, focusing on aspects of fish welfare not previously addressed in Korea. By reviewing existing research on international standards for fish welfare, we assessed the welfare criteria essential for farmed fish [40,41,42,43]. Like terrestrial animals, farmed fish should be entitled to the Five Freedoms [40,41,44,45], ensuring freedom from pain and discomfort throughout production and transport [43]. Therefore, we identified elements within the aquaculture process that could cause pain and distress in fish. 

Changes in water quality were identified as the most significant factor contributing to olive flounder mortality, with 84.5% of respondents acknowledging its impact. The chi-squared test confirmed a significant difference between Wando and Jeju-do (χ^2^ = 15.3526, *p* = 0.0015). This suggests that regional differences in water quality management practices or environmental conditions significantly affect fish mortality rates. These results emphasize the need for region-specific water quality monitoring and management strategies to mitigate impacts on fish health.

High and low water temperatures were also critical factors, affecting 81.7% of respondents. The significant chi-squared result (χ^2^ = 14.9386, *p* = 0.0019) indicates that temperature management practices or climatic conditions differ significantly between the two regions. This highlights the importance of implementing adaptive management practices to cope with temperature fluctuations, possibly incorporating advanced technologies for real-time monitoring and control of water temperatures. 

While 80.3% of respondents identified parasites and viruses as significant mortality factors, the chi-squared analysis did not reveal a significant difference between the regions (χ^2^ = 3.9422, *p* = 0.2678). This suggests that both Wando and Jeju-do face similar challenges in terms of parasite and virus management. Consistent strategies such as regular health checks, vaccination programs, and improved biosecurity measures should be emphasized to address these issues in both regions effectively. 

Feed management emerged as a critical area with significant regional differences. The chi-squared statistic (χ^2^ = 44.9948, *p* < 0.001) indicates significant differences in feed practices between Wando and Jeju-do. While 69% of respondents identified overfeeding or spoiled feed as a mortality factor, 97% kept records of feed quality and all managed temperatures to prevent spoilage. Despite these efforts, overfeeding and feeding spoiled feed remain common problems [31]. In addition, our survey revealed a considerable preference for moist pellets (MPs) over formulated feeds. The preference for MPs is attributed to their ability to promote rapid growth and superior fat accumulation in fish, thereby commanding higher market prices. However, MP production relies on low-grade or juvenile fish captured offshore and along the coast as the primary ingredient, lacking an established management system for self-made feeds, including quality control and feed testing [46,47,48,49,50,51]. However, there are currently no standards for managing MPs as a form of raw fish feed. These findings highlight the need for stricter feed management protocols, regular training of farmers on optimal feeding practices, and the introduction of more robust quality control measures.

Medication use showed significant regional variation (χ^2^ = 22.1591, *p* < 0.001). The frequent use of medication, especially in Wando, reflects a proactive approach to disease management. However, the variable responses from Jeju-do indicate inconsistent practices. Standardizing disease management protocols, improving access to veterinary services, and promoting the use of effective and sustainable disease control measures are essential to ensure consistent and effective disease management in both regions [31,52,53].

Handling and transport practices were critical, with all respondents emphasizing the importance of minimizing physical damage and stress management [35,54]. The chi-squared test for injury during transport (χ^2^ = 59.6859, *p* < 0.001) revealed significant regional differences, with all Wando respondents citing transport as a cause of injury compared with 88.6% of Jeju-do respondents who reported no injuries. This discrepancy highlights the need for improved transport practices, including better handling techniques, improved transport equipment, and stricter regulations to ensure the welfare of fish during transport.

Staff training is essential to maintain high standards of aquaculture practice. The survey results reveal significant regional differences in training practices between Wando and Jeju-do. A total of 71.4% of respondents reported sporadic training in Jeju-do, while in Wando, 91.7% had regular training 12 times a year or less. The chi-squared test (χ^2^ = 11.29, *p* = 0.0234) confirms a significant difference in training provision between the regions. Overall, 83.1% provided training, but the frequency and duration varied. Training in Jeju-do was sporadic, leading to potential knowledge gaps, while training in Wando was regular but shorter. Effective training is essential for high standards of aquaculture. Tailored, comprehensive programs are needed to address these regional gaps. Such programs should address specific regional needs and ensure that all farmers are well prepared [38,55]. Addressing these disparities will improve fish welfare and productivity in aquaculture.

Awareness of fish welfare varied, with 47.2% of Wando respondents and 34.3% of Jeju-do respondents reporting awareness, although this difference was not statistically significant (χ^2^ = 0.752, *p* > 0.05). However, attitudes toward moving to welfare-oriented practices showed significant regional differences (χ^2^ = 26.92, *p* < 0.05), with only 32.4% of respondents willing to adopt such practices. This reluctance, mainly due to “lack of interest” and “perceived need”, suggests a critical need for educational initiatives and policy adjustments. Promoting the benefits of welfare-oriented practices through targeted education and incentives can help to encourage wider adoption and implementation, ultimately improving the sustainability and productivity of aquaculture.

However, the study had several limitations. Due to the lack of previous research on fish welfare in Korea and the lack of available welfare criteria, we relied on interviews, surveys, and site visits, potentially leading to biased or incomplete conclusions and affecting the generalizability of the findings. In addition, we focused only on olive flounder aquaculture facilities in Jeju-do and Wando, which may have overlooked differences in welfare practices and challenges across different fish species and geographical regions in Korea. Future studies should, therefore, consider diversifying the species farmed and expanding the survey to include more aquaculture facilities nationwide. Addressing welfare issues associated with other commonly farmed fish species in Korea is imperative. It is necessary to investigate the awareness of fish farmers and consumers regarding fish welfare and to conduct scientific studies on the relationship between fish welfare and stress. Finally, the development of species-specific fish welfare criteria and assessment methods can facilitate the establishment of guidelines and protocols relevant to fish farming. These initiatives and actions will help raise the awareness of fish welfare within the aquaculture industry.

## 5. Conclusions

Despite its high production volume, the neglect of fish welfare in Korean aquaculture raises significant concerns (see Table 2). The current emphasis on quantitative production methods has resulted in various welfare-related challenges and risks, threatening the sustainability of the industry and potentially compromising the quality of the fish produced. Unfortunately, established guidelines for fish welfare in Korea are lacking, and relevant studies addressing these issues based on appropriate welfare criteria for domestically reared species are absent. This study, primarily centered on olive flounder, draws upon prior overseas research and field surveys to analyze welfare issues. To ensure the long-term sustainability of aquaculture, scientific studies should be conducted to establish distinct welfare criteria for different species and explore practical methods to implement these criteria effectively in aquaculture facilities.

## Figures and Tables

**Figure 1 animals-14-01693-f001:**
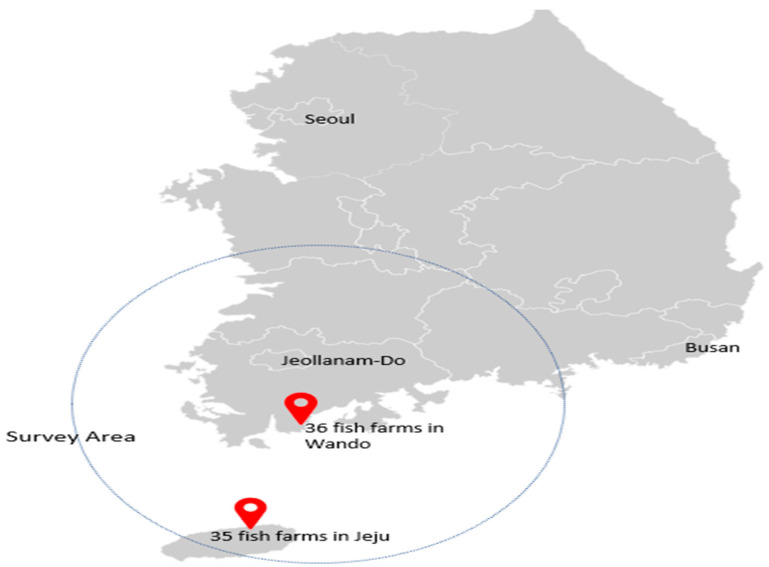
Survey area. Source: drawn by authors.

**Table 1 animals-14-01693-t001:** Characteristics of survey respondents involved in olive flounder farming.

Basic Questionnaire Components		Number of Respondents	Ratio (%)
Location of fish farms	Wando	36	50.7
	Jeju-do	35	49.3
Years of experience	Less than 10 years	14	19.7
	11–20 years	15	21.1
	More than 20 years	42	59.2
Involvement in the operation of the fish farm	Yes	69	97.2
	No	2	2.8
Facilities area (m^2^)	Less than 4000	38	53.5
	More than 4000	33	46.5
Water surface area (m^2^)	Less than 3000	33	46.5
	More than 3000	38	53.5

**Table 2 animals-14-01693-t002:** Welfare issues identified in olive flounder fish farms through surveys.

Management Factors of Fish Farms	Issues and Risks Related to Fish Welfare
Water quality	Difficulties in managing water quality and temperature changes;
Feed management	Unregulated use of MPs predominates;
Disease management	Lack of guidelines for antibiotic use and medication baths, with self-diagnosis leading to potential misuse of medications;
Transportation	Long-distance transportation in narrow tanks at high densities
Training	Lack of systematic education on fish management.

Source: The author summarized the results of the survey.

## Data Availability

Raw data are not publicly available or stored elsewhere because of ethical and privacy issues. Some anonymous data collected in this study can be requested from the first author, although its availability will require the participants’ consent.

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
