# Peer review of "Fish Welfare-Related Issues and Their Relevance in Land-Based Olive Flounder (Paralichthys olivaceus) Farms in Korea"

_animals, 2024, doi:10.3390/ani14111693_

Round 1
Reviewer 1 Report
Comments and Suggestions for Authors
Dear authors,
Thank you for your contribution with this manuscript which, is not only important to shed light on the welfare of Korean fish farms but also can create guidelines for these animals.
In general, the manuscript has a great lack of information, especially in the methods section. There’s a lot of information that it’s in the wrong section, e.g. there’s a lot of explanation in the results section that it’s not clear that comes from the results of the manuscript. The results also need to be improved either by removing most of the text that is not results, but also by adding more information on the results themselves. The discussion has more information about the results than the discussion about it. I would suggest a careful modification of the manuscript as, as it is, I’m afraid that it’s prepared to be accepted.
Recommended modifications:
Materials and Methods:
The authors started this section with their objectives. This should be moved to the end of the introduction as this section should only mention the materials and methods used to do the project.
The authors mentioned that they performed surveys. What kind of questions were present in the surveys? The authors add an Appendix A in the end of the manuscript but did not mention it through the methods. Was the questionnaire done by the authors? Or adapted from some source? Appendix A is for face-to-face interviews, what about the online ones? Was the online questionnaire done with multiple choice questions or descriptive ones as the face-to-face ones? If the responses were all descriptives, how the authors evaluate the perception of the participants? The authors mentioned the number of facilities, but is it the same as the number of questionnaires? How many questionnaires were analysed? Were all replied by the participants? Were there questionnaires that were not completed filled? Was the questionnaire validated by any assessment? All this information must be included in the methods.
Regarding the participants, which ages are we talking about? Are all the participants in the same age range? With the same expertise years? Are all in the same position within the fish farm? These details must be included so the reader knows what type of experimental group we are talking about.
What statistical analysis did the authors use? There’s no information about it apart from the confidence level. How did the authors analyse the data? This must be included in detail in this section.
Results:
I was a bit confused by section 3.1 as it does not show results, but instead the description of the aquaculture in Korea. This section should be moved to Methods or even Introduction as they do not show results from the manuscript.
Line 137: Did the authors check the paper regarding stocking density from Saraiva et al., 2022 (Finding the “golden stocking density”: a balance between fish welfare and farmers' perspectives. Frontiers in Veterinary Science, 9, 930221.)? Could also be a good information source or to add to the paper in the discussion if the authors think it makes sense. This comment is just a suggestion.
Line 139-145: I feel that this is a discussion and not results, so it should be moved to the discussion section.
Same for sections 3.2.2 to 3.2.5. Are these sections related to the surveys done by the authors? If yes, it should be made clear and remove everything that does not belong to the Results section. If not, it should not be in this section at all. This section should address only data from the current project/manuscript and not results or information from others. It is also addressed in this section some suggested actions by the authors which should definitely be moved to the discussion.
Line 198-203: These lines belong to the Methods section.
Line 206-208: These lines should go to Introduction or discussion.
Results actually start only in line 209 with the sentence “Survey results revealed…”
Line 225-228: Should be moved to discussion as it’s not results but speculation from the authors to justify this result.
Line 228: “On a positive note…” Results are not positive or negative, are results. This type of sentence should be moved to discussion.
Line 234-238: This is discussion. Again, the authors should not suggest anything in the Results section, there’s a Discussion section for that.
Discussion:
Line 273: Authors mention “…the utilization of growth promoters was minimal…” although line 219 says “However, 93% of the respondents indicated that some farms prioritized MP over formulated feed.” and in line 149 they describe that “The preference for MP stems from its ability to promote rapid growth…” and then in line 280 mention that “However, according to the survey, the use of MP was far higher than that of formulated feed.” In this sense, I’m confused about the information the authors want to send to the reader.
Line 283: “Equipment management for disease prevention and hygiene, seed procurement, and pharmaceutical record-keeping were generally satisfactory.” How can authors say that was satisfactory? What is the cutting value to assume something is satisfactory, especially for the pharmaceutical data? Are there any references authors can add to this sentence to confirm their sentence?
Line 312: “The survey conducted had a simple margin of error of ±11.63% with a 95% confidence level, which was considered acceptable.” But what this really mean? If authors don’t explain what they did statistically this comes with zero information to the reader.
Line 326: Where did Table 3 come from? It just shows in the discussion with no indication in the text.
Author Response
[2 May 2024]
Animals
Dear Editor and Reviewer:
We/I wish to re-submit the manuscript titled “Fish welfare-related issues and their relevance in land-based olive flounder (Paralichthys olivaceus) farms in Korea.” The manuscript ID is 2945233. _______.
The manuscript has been rechecked and appropriate changes have been made in accordance with the reviewers’ suggestions. The responses to their comments have been prepared and attached.
We thank you and the reviewers for your thoughtful suggestions and insights, which have enriched the manuscript and produced a better and more balanced account of the research. We hope that the revised manuscript is now suitable for publication in your journal.
Thank you for your consideration. I look forward to hearing from you.
Sincerely,
Seunghyung Lee
Major of Aquaculture and Applied Life Sciences,
Division of Fisheries Life Sciences, Pukyong National University,
Busan, Korea
Correspondence: shlee@pknu.ac.kr

Reviewer 2 Report
Comments and Suggestions for Authors
The article proposed by Seoyeon Oh and Seunghyung Lee concerns aquaculture in Korea, specifically the olive flounder (Paralichthys olivaceus) land-based farms. The study is based on interviews and surveys conducted on olive flounder farms in Jeju Island and Wando. The authors assessed issues affecting fish welfare, including stressors, mass mortality events, and disease outbreaks, which significantly challenge the welfare of farmed fish. Furthermore, this study highlighted farmers' diverse perceptions of fish welfare and emphasised the need for a common and innovative approach. Although the article provides interesting insights, the authors must improve some aspects of the manuscript. The language comes across as cumbersome and is almost developed through machine translation. In the manuscript, there are many repetitions (es. IT IS IMPERATIVE – FISH SUFFERING AND PAIN- FISH WELFARE- FISH STRESS), so it is recommended to rewrite some sentences to improve the form, avoid repetitions and add technical information. I recommend the publication of the manuscript with only minor corrections. Please find more specific comments below.
SIMPLE SUMMARY
Line 10-11 – which factors?
Line 11-12- please insert the comma missing as shown: elevating fish welfare not only enhances the sustainability of aquaculture, but also fortifies fish resilience against stressors.
Line 12-13 please modify the sentence as shown: Despite significant advances in production and species diversity in Korean aquaculture, the welfare of farmed fish remains largely neglected.
Line 14-15 please modify the sentence as shown: To identify potential welfare issues within the farming process,
line 18 modify the sentence structure as shown -- in recent decades Korean aquaculture has witnessed remarkable expansion;
line 19-20 please modify the sentence as shown :We analysed aquaculture practices of the olive flounder, the predominant species in Korean consumption and production, through the lens of fish welfare.
Line 23-24 please modify the sentence as shown: These problems have involved stressors, mass mortality events and disease outbreaks, putting a strain on the welfare of farmed fish.
Line 24-25 please modify the sentence as shown: In addition, our survey highlighted the heterogeneity of fish welfare perceptions among farmers, underscoring the need for a cohesive approach.
Line 29-30 please modify the expression as shown: strengthen its growth
INTRODUCTION
Line 38-40 please insert the commas missing as shown-- They possess sensory receptors sensitive to pain and manifest physiological and behavioural stress responses, analogous to mammals, when subjected to painful handling procedures
Line 44 please modify the expression as shown: considers it essential
Line 45 please modify the expression as shown: increasingly recognized
Line 61-64 please insert the commas missing as shown -- Prioritizing aquaculture methodologies, that meticulously account for fish stress and suffering, constitutes a pivotal stride toward addressing this concern, thereby safeguarding farmed fish welfare and curbing adverse environmental impacts.
Line 74 -- please modify the sentence as shown: assessing farmers' awareness of fish welfare
Line 75 – please modify the expression as shown: by analyzing
RESULTS
From 3.1 to 3.3 Please provide more technical information and add related bibliographic references to support what is written. These chapters require a deep insight into the main topics according to available bibliographic data.
Line 112-113 – imprecise – if you want to say many others, you say it at the end of the sentence, not in the middle. Please specify
Line 125 – redundant – you repeat the same thing: several critical areas of concern… several critical areas.
Line 130 - please modify the sentence as shown: we support the implementation of best
Line 135 - please modify the expression: an overabundance of fish in tanks with an excess
Line 139- please modify the expression: distributed across multiple tanks with spread
Line 141-142 please modify the expression as shown: and this may require the use of antibiotics to treat or prevent injury.
Line 142-145 please modify the expression as shown: In addition, some land-based fish farms, particularly those in unsuitable locations such as Jeju Island, lack systematic water management practices, making them susceptible to the spread of pathogens and parasites, further compromising fish welfare.
Line 160- please modify the expression: Although fish farms employ diverse methods with different methods
Line 164 -- please modify the expression: can impede accurate diagnosis with can prevent
Line 170 - please modify the expression: such as olive with such as the olive
Line 177 - please modify the expression: may be suboptimal with may not be optimal
Line 181 please modify the expression: needless pain ---not proper in a scientific article. You should not provide personal opinions or you have to say it for every trauma of the fish.
Line 183- please modify the expression: during transportation and prior to consumption with and before consumption
Line 191 -- please modify the expression: feed and feed organisms--repetition –wrong translation
Line 195 - please modify the expression as shown: Consequently, it is essential to establish specific guidelines
Please revision each line where is written it is imperative and change it with it is essential
Line 201- 203 the concept that To obtain additional information, a survey of 71 fish farms was conducted, including 36 in Wando and 35 in Jeju. Is already said in line 95
Line 227-- please modify the expression as shown: concerns about abuse
Line 234- please modify the expression: responded with answered
Line 235-238- many repetitions, please modify the expression as shown -Thus, the fish farmers tend to rely on their experiences and knowledge to ensure safe transportation. These findings suggest the need to improve fish welfare (transportation) guidelines, highlighting an area for improvement in fish welfare practices.
Line 242-245 please insert the commas missing as shown --Further investigation through in-depth interviews with researchers and academics revealed that although some training on aquaculture practices is provided within the facility, it may not be effectively implemented to incorporate factors such as fish stress and welfare.
DISCUSSION
LINE 291-- please modify the expression: the imperative to enhance awareness with the need to
Line 299- please modify the expression: potential misuse with improper use
Line 303- please modify the expression: survey analysis unveiled additional administrative with revealed
Line 324 - please modify the expression: endeavors with efforts
CONCLUSIONS
Line 338- please modify the expression: diverse welfare criteria with different
Appendix A- the questionnaire has little scientific value, most of the questions are personally motivated, and the answer may not be truth, unreliable.
Line 347 – please define fingerlings
Comments on the Quality of English LanguageModerate editing of English language required
Author Response

(The authors gave the same response as above.)

Round 2
Reviewer 1 Report
Comments and Suggestions for Authors
Dear authors,
I hope this message finds you well.
Although there were some improvements in the manuscript. Some text that should be deleted (and was in the report that was deleted/moved) is still in the manuscript. I don’t know if it’s some problem with the file but the manuscript to download is still showing these lines of text.
Authors should still add statistical analysis to their data. I mentioned in more detail below.
I hope to receive the next version of the manuscript with some statistical analysis and not only percentages.
Comments by line:
Line 109: I think note #2 is not necessary as it is already in the main text.
Line 118: The source of the table is not necessary. When it’s not cited it’s assumed it’s the author's source of information.
Line 128: It will be necessary to add the source for the figure, assuming it was not created by the authors.
Line 132: When authors say [see 3.1] is related to what? If is section 3.1, it should not appear in the text as it’s an upcoming section. If it’s something else, please clarify.
Line 149: Same comment as line 118.
All sections 3.1 (Characteristics of Aquaculture in Korea), 3.2 (Fish Welfare Issues at Land-Based Olive Flounder Farms in Korea) and all sub-sections, and 3.3 (Fish Farmers' Perception of the Welfare of Olive Flounder) must be removed from the current location as Results should not include this kind of text, as already expressed in the last revision comments. The characteristics of aquaculture in Korea should be in the introduction and not results unless it comes directly from the data, which it does not seem that way. In the results, only section 3.2 (3.2. Survey findings and farmer perspectives) should be present.
Line 169-178: This should be moved to discussion as already pointed in the last report.
Results: The authors continue to have no statistical analysis for their study. Even though it seems linear to say from the responses that one variable is “…were the most significant factors contributing to mortality” they cannot say that because they have no statistical analysis. The authors replied to my previous comment with “As a limitation of this study, there is currently no research on fish welfare in Korea, so no data is available.” However, the statistical analysis for their data has nothing to do with the absence of data from the topic. The authors have a questionnaire and that’s their data where they can apply statistical analysis. I highly recommend that the authors check how to perform data analysis on questionnaires so they can actually say which factors are most significant. They can perform a PCA (Principal component analysis) on their data and report the significance. Just the percentages cannot say anything regarding the significance of the data. Authors can be reporting something that it’s not significant (especially that they don’t even present the standard deviation for the findings).
Line 283-289: This is a discussion and should be moved to the appropriate section, as reported previously. Also, Results have no positive note, as also reported.
Table 3 should be included in the results. A table should be cited in discussion for the first time only in very rare exceptions. I don’t think this is the case.
Line 441: I’m assuming this is from Table 3 and it should be deleted.
Author Response
Dear Editor and Reviewer:
I would like to thank you and the reviewers very much for your constructive suggestions on our manuscript animals-2945233.
The manuscript has been rechecked and appropriate changes have been made in accordance with the reviewers’ suggestions. The responses to their comments have been prepared and attached.
We thank you and the reviewers for your thoughtful suggestions and insights, which have enriched the manuscript and produced a better and more balanced account of the research. We hope that the revised manuscript is now suitable for publication in your journal.
I would like to mention that there seems to be an error in the MDPI paper upload system. In the file I uploaded, the sentence I deleted as per the reviewer's comment was not removed.
Thank you for your consideration.
I look forward to hearing from you.
Sincerely,
Seunghyung Lee
Major of Aquaculture and Applied Life Sciences,
Division of Fisheries Life Sciences, Pukyong National University,
Busan, Korea
Correspondence: shlee@pknu.ac.kr
